# Structural Insight and Development of EGFR Tyrosine Kinase Inhibitors

**DOI:** 10.3390/molecules27030819

**Published:** 2022-01-26

**Authors:** Tasia Amelia, Rahmana Emran Kartasasmita, Tomohiko Ohwada, Daryono Hadi Tjahjono

**Affiliations:** 1School of Pharmacy, Bandung Institute of Technology, Jalan Ganesha 10, Bandung 40132, Indonesia; tasiaamelia@fa.itb.ac.id (T.A.); kartasasmita@fa.itb.ac.id (R.E.K.); 2Graduate School of Pharmaceutical Sciences, The University of Tokyo, Tokyo 113-0033, Japan; ohwada@mol.f.u-tokyo.ac.jp

**Keywords:** activation, binding, EGFR, inhibitor, kinase

## Abstract

Lung cancer has a high prevalence, with a growing number of new cases and mortality every year. Furthermore, the survival rate of patients with non-small-cell lung carcinoma (NSCLC) is still quite low in the majority of cases. Despite the use of conventional therapy such as tyrosine kinase inhibitor for Epidermal Growth Factor Receptor (EGFR), which is highly expressed in most NSCLC cases, there was still no substantial improvement in patient survival. This is due to the drug’s ineffectiveness and high rate of resistance among individuals with mutant EGFR. Therefore, the development of new inhibitors is urgently needed. Understanding the EGFR structure, including its kinase domain and other parts of the protein, and its activation mechanism can accelerate the discovery of novel compounds targeting this protein. This study described the structure of the extracellular, transmembrane, and intracellular domains of EGFR. This was carried out along with identifying the binding pose of commercially available inhibitors in the ATP-binding and allosteric sites, thereby clarifying the research gaps that can be filled. The binding mechanism of inhibitors that have been used clinically was also explained, thereby aiding the structure-based development of new drugs.

## 1. Introduction

Epidermal Growth Factor Receptor (EGFR) is one of the receptor tyrosine kinases (RTKs), members of the ErbB/HER family which consists of ErbB1 (EGFR or HER1), ErbB2 (HER2 or Neu), ErbB3 (HER3), and ErbB4 (HER4). They share similar basic structures such as an extracellular ligand binding, an α-helix transmembrane, a cytoplasmic tyrosine kinase (except ErbB3), and carboxy-terminal signaling domains [1]. The growth factors of the ErbB family are divided into three groups, namely the specific ligand of ErbB1 or EGFR (such as EGF, TGF-α, and amphiregulin), those that bind to ErbB1 and ErbB4 (such as heparin-binding EGF, betacellulin, and epiregulin), and ErbB3/ErbB4 (such as neuregulins and heregulins) [1]. However, unlike the others, ErbB2 has no specific ligand. It binds to any ligand similar to the one that can activate it and form a dimer [2].The ligand-binding process induces homodimerization and heterodimerization with other ErbB family members and activates the tyrosine kinase domain [1]. The activation of this intracellular domain causes autophosphorylation and enables it to interact with signaling components to downstream signaling pathways. Furthermore, this was performed through the RAS-RAF-MEK-MAPK, PI3KPTEN-AKT, and STAT pathways [3,4] to enhance cell proliferation and inhibit apoptosis (Figure 1) [5].

EGFR signaling was observed in most lung cancer cases. Overexpression of this receptor has been identified in 40 to 89% of NSCLC cases, with the highest and lowest rates in squamous tumors and adenocarcinoma [6,7]. Besides its high rate in NSCLC, the level of EGFR activation is also related to bad prognosis and tumor regression rate [8]. The phosphorylation activity in the kinase domain of EGFR may also be irregulated by several mechanisms, including EGFR mutation and overexpression, which is commonly found in tumor cells [9]. This indicates that improper activation of tyrosine kinase promotes tumor progression and inhibits cell apoptosis [10]. Spontaneous oligomerization was also reported in several studies for mutated EGFR which lacked part of residues in the extracellular domain and induced autophosphorylation [11,12].

EGFR also interacts with the integrin pathway and triggers matrix metalloproteinases to alter and stimulate cell adhesion, motility and invasion, and promote metastases [13,14,15]. Furthermore, the overexpression or enhanced activation of EGFR mutations occurs in several NSCLC cases, leading to constitutive TK activity. This makes it a rational target for therapeutic intervention and also promotes the development of novel anticancer agents targeting EGFR.

## 2. Structure of EGFR

### 2.1. Extracellular Domain

The human EGFR encodes 1210 amino acids with a molecular mass of approximately 134 kDa [16]. It is located in the 7p12-14 region of chromosome 7, which consist of 28 exons (Figure 2a) [17]. The first 24 amino acids are the signal peptide of this protein that are often excluded in the structural numbering.

The extracellular ligand-binding domain of EGFR contains 620 amino acids (25-645) which are divided into four subdomains, namely I (L1), II (CR1), III (L2), and IV (CR2). L1 and L2 are both leucine-rich domains with a β-helix structure, and are responsible for growth-factor binding [19]. Both CR1 and CR2 are cysteine-rich regions with disulfide bonds. CR1 plays a role in the homo- or hetero-dimer formation of EGFR with other ErbB family members [19,20,21]. L1, CR1, and L2 form a C shape which accommodate EGF to bind between L1 and L2. Based on the interaction of EGFR in the binding site, three sites were defined in both subdomains (Figure 3). The B loop of EGF interacts with site 1 of L1 by hydrophobic interaction with Leu14, Tyr45, Leu69, and Leu98, and hydrogen bonds to residue 16 to 18 which forms a parallel β-sheet. The A loop of EGF hydrophobically interacts with Val350 and Phe357 on site 2 of L2, while Arg41 of EGF forms a salt bridge with Asp355 in L2. The C-terminal region of EGF forms a hydrophobic interaction with Leu382, Phe412, and Ile438 as third site of L2. The residues Gln43 and Arg45 are also able to form hydrogen bonds with the side chain of Gln384. Furthermore, in the dimeric EGF–EGFR complex, the two EGF ligands are located on the opposite side of the dimer which are 79 Å apart from each other [21].

### 2.2. Transmembrane Domain

The transmembrane (TM) domain consists of ~22 amino acids (646–668) which connect the extracellular and intracellular domain of EGFR. Previous studies revealed the critical role of the TM domain in the allosteric modulation of EGFR by two activation pathways that involve pivoting and rotational motion of the TM helices [22,23]. The three-dimensional structure of the TM domain in the presence of the juxtamembrane domain shows interhelix contact at the N-terminal small-X3-small motif, which is experimentally stable [22,24]. Furthermore, the lipid environment of TM helix dimers influences the stability of certain conformations, as shown in the Coarse-Grained MetaDynamics (CG-MetaD) simulation conducted by Lelimousin et al. (2016). It was found that the thickness of the bilayer can determine the motions of the TM helix, thereby affecting the stability and receptor activation [22,25,26,27,28,29].

### 2.3. Juxtamembrane Domain

This links the C-terminus of the TM domain to the kinase domain of EGFR and plays important role in its dimerization and activation [30,31,32]. The juxtamembrane domain is a flexible region that is often absent from crystal structures. Furthermore, interdisciplinary approaches using X-ray crystallography and NMR were successful in determining high-resolution JM domain structures which were also useful in silico study [33,34]. Even though it is a short region in EGFR, containing only ~37 residues, it consists of lysosomal and basolateral sorting motifs [35,36], a nuclear localization sequence [37], calmodulin-binding site [38], protein kinase C [39,40], and MAPK phosphorylation sites [32,41]. A study revealed that mutation of Thr654 within the JM domain caused an increase in kinase phosphorylation. It was also found that the phosphorylation of Thr-654 diminishes the ability of this region to modulate the receptor activation [32].

### 2.4. Kinase Domain

The EGFR tyrosine kinase (TK) domain consists of an NH_2_-terminal lobe (N-lobe) which comprises five β-sheet strands (β1-5) and one αC-helix spanning from residue 729 to 744, and a larger COOH-terminal lobe (C-lobe) comprising five α helices (αE, αF, αG, αH, and αI). The ATP-binding site is located in a cleft between the two lobes, beneath a highly conserved glycine-rich phosphate-binding loop that links β1 and β2 in the N-lobe [42]. The glycine-rich loop coordinates closely with the phosphates of ATP via backbone interactions [43].

In the active state, the conserved glutamate Glu738 in the αC helix forms an ion pair with Lys721 in β3 that interacts with the phosphate groups of ATP. The C-lobe surrounds the ATP-binding cleft from below and contributes to a highly conserved catalytic loop (Asp812-Asn818). Furthermore, the Asp812 interacts with the attacking hydroxyl side chain of the tyrosine substrate, while Asn818 forms hydrogen bond interactions that orient Asp812. The C-lobe also contributes to the regulatory activation loop Asp831-Val852 which has a conserved Asp-Phe-Gly (DFG) motif as its base [44].

### 2.5. C-Terminal

According to kinetic studies, the C-terminal domain of EGFR consists of 229 amino acids (residue 982–1210) and plays an important role in regulating receptor activation by suppressing kinase activity in the absence of autophosphorylation [45,46]. This domain is a proline-rich residue that has phosphorylation sites [1]. Due to its flexible structure, it is often disordered in the published crystal structures of EGFR, especially the polypeptide chain (residue 990–1005) which is highly fluctuated in MD simulation [47]. It has a proximal tail that is important in the autoinhibition activity of the receptor and has been studied structurally [48,49]. Residue 997–1001 forms an α-helix structure called AP-2 helix because it interacts with clathrin-associated protein complex AP-2 [50]. By interacting with the N-lobe of the second kinase in the dimer, this helix maintains two kinase domains in an inactive dimer [51,52]. The AP-2 helix is followed by a hook spanning from residue 1003 to 1022, which are acidic and interact with the hinge region of the kinase. This hook mediates an inhibitory dimerization of the receptor by electrostatic interaction, thus it is called an electrostatic hook. However, these interactions are destabilized by phosphorylation [49]. The last part of this hook forms a β-strand that prevents the formation of the JM latch [48]. The deletion of the distal residue Tyr1210, which is also part of the NPXY motif, significantly reduces the phosphorylation of Tyr869 in the kinase domain [48]. Therefore, these factors conclude that Tyr1210 is important for kinase activation.

## 3. Active–Inactive Conformation

Kinase is an enzyme that catalyzes phosphorylation processes. It facilitates the transfer of phosphate groups from a phosphate donor such as ATP to a specific substrate. There are more than 500 protein kinases encoded by the human genome, which have a highly conserved catalytic domain formed by α-helix C-terminal and β-strand N-terminal lobes as ATP-binding sites [53]. The activation loop within those kinases contains tyrosine, threonine, or serine that is phosphorylated and regulates kinase activity [54]. Two regions are commonly used to indicate the confirmation of active and inactive kinase, they are the αC helix and DFG motif in the activation segment. Furthermore, the kinase and αC helix located at residue 753–767 in EGFR are twisted inward against the N-lobe and towards the active site. This conformation shortens the distance between Glu762 of αC helix and Lys745 of β3 strands, allowing a salt-bridge formation and further interaction with α- and β-phosphate groups of ATP [55].

The activation segment of EGFR is located at residue 855-884. The Asp-Phe-Gly (DFG) motif at the beginning of the activation loop plays an important role in protein catalysis [56]. This DFG motif shows conformation in the active kinases where the aspartate is pointing to the ATP-binding site, allowing the magnesium ion to bind to the β- and γ- phosphate groups of ATP, which are known as DFG-*in* conformation [57]. However, this DFG motif flips outward in the inactive kinases, causing the aspartate to no longer coordinate the magnesium ion at the catalytic site, which is further known as DFG-*out* conformation [58]. An alteration of aspartate and phenylalanine orientation is found in this *out* form, where phenylalanine occupies the in-position of aspartate [55]. Based on the mutagenesis study, a residue was proposed to influence the sensitivity of kinases towards inhibitors that stabilized DFG-out conformation. It acts as “gatekeeper” since it has a bulky size and occupies the position that prevents ATP or other molecules from accessing the inner hydrophobic pocket [59]. Moreover, the “gatekeeper” residue Tyr790 mutation to methionine has been known to cause resistance to inhibitors due to the enhanced affinity for ATP [60].

In a study by Zhao et al., (2019), EGFR kinase conformations are classified into six classes which consist of two classes of active conformation and four classes of inactive conformation (Figure 4). In all classes, the side chain of aspartate has three different positions, namely DFG-*in*, DFG-*out*, and DFG-½*in*. Unlike the others, the side chain of Asp points to the upside [61]. The αC helix also shows three states, namely *in*, *out*, and in between or ½*out*. The first two classes of active kinase have DFG-*in* conformation with the αC helix in the *in* and ½*out* states. In the inactive EGFR, the DFG motifs have three different positions, while all the αC helices are in the *out* positions [62]. These various conformations of the EGFR kinase binding site provide essential information for all possible sites outside the ATP-binding site in order to be a benchmark for the discovery of novel inhibitors. The binding pocket sizes of Class 1-5 were not significantly different and ranged from 950 to 1119 Å^3^. However, the volume was significantly increased in Class-6 conformation to 1913 Å^3^, allowing it to bind with various ligands [62].

There are other common, major regions in the kinase receptors beside the αC helix and activation loop, namely the N-lobe, ATP binding and allosteric sites, and the hinge region. The N-lobe consists of one α-helix and five antiparallel β-sheets, and it is connected to the C-lobe via a hinge region in the ATP-binding site [67]. The ligand binding causes conformational changes in this protein structure and enables autophosphorylation. In the structural view, the C-lobe also shrinks and moves closer to the N-lobe during phosphorylation and activates the kinase receptor [68].

## 4. Mutation of EGFR and Resistance Mechanism

Somatic mutations of EGFR tyrosine kinase domain in NSCLC patients were first reported in 2004, in which there was a deletion in exon 19 and point mutations in exon 21 [6,69]. According to the type of nucleotide changes, the EGFR mutations are grouped into three classes. Class I mutations involve short deletions that result in the loss of four to six residues between E746 to S752 encoded by exon 19. Class II mutations involve substitution of a single residue which occurs between exon 18 to 21. Class III mutations include duplications or insertions that mostly occur in exon 20 [70,71]. Most tyrosine kinase domain mutations (85–90%) had a deletion in exon 19 and substitution of L858R in exon 21. Recent studies have also reported a rare exon 22 mutation (E884K) that may reduce sensitivity to different EGFR inhibitors [72]. Several mutations at the cytoplasmic region cause destabilization of the conformation, upregulation of kinase activity, and irregularly promote downstream signaling pathways by avoiding cell apoptosis [73,74,75].

The mutated EGFR shows a resistance mechanism to the inhibitor used clinically and limited drug efficacy. This was found in lung cancer patients with substitution of threonine in the position of 790 to methionine (T790M) [76,77,78]. Half of all cases of resistance to the first-generation EGFR kinase inhibitor were caused by the T790M mutation [79,80]. Thr790 is also referred to as a “gatekeeper” residue because it is located in the entrance of the hydrophobic pocket of the ATP-binding pocket which determines the specificity of the inhibitor in the kinases. Substitution of Thr790 to a bulkier residue such as methionine causes steric hindrance for inhibitor binding [76,77,78].

Though this mutation had resistance to gefitinib and erlotinib, it is still sensitive to irreversible inhibitors such as EKB-569 and HKI-272 [77,78,81,82]. Despite the fact they share the same quinazoline core and aniline group as the first-generation inhibitors, they also contain a crotonamide as a Michael-acceptor group that is able to form a covalent bond with Cys797 at the ATP-binding site [60]. This prompted the investigation into why the irreversible inhibitors are able to interact with the binding pocket even though it has an aniline group that caused steric hindrance for gefitinib. However, this has been discovered by a study that revealed the resistance mechanism of this secondary mutation. The T790M substitution increases the ATP affinity to the binding pocket and allows it to compete with the inhibitors [BioRender object] resulting in a resistance mechanism [60].

The second mutation occurred in about half of the patients that were treated with EGFR-TKIs, despite the fact that occurrence before therapy was uncommon. Furthermore, the T790M mutation is also found in combination with the others that mostly coexist with an L858R point mutation [83,84]. The double mutation L858R/T790M activates EGFR and reduces ATP’s affinity to the binding site at the same time. The L858R/T790M mutation is more resistant to EGFR TKIs compared to the single L858R mutation or exon-19 deletion in the NSCLC cell line [76,85,86].

The other non-T790M mutation that was first discovered is D761Y, which commonly coexists with an L858R mutation. The L858R/D761Y mutation was more resistant to gefitinib compared to the cell with L858R alone, although it was still less resistant than L858R/T790M. This mutation was found to be less sensitive to the irreversible inhibitor, HKI-272, compared to the gefitinib response, which suggests that different mutations of EGFR will cause different responses to the reversible and irreversible inhibitors [80]. Though the resistance mechanism of this mutation is still unclear, it induces a conformational change between active and inactive states of EGFR and affects the binding ability of the inhibitor [80,87]. Recently, an L858R/T854A mutation was also discovered in a NSCLC patient that received long-term treatment of first-generation TKIs [88]. This mutation was resistant to erlotinib, showing a 3-fold increase in erlotinib concentration compared to the L858R single mutation. Furthermore, L858R/T854A was still more sensitive to erlotinib than L858R/T790M by more than 300-fold, and had more resistance than L858R [89]. Recently, a list of rare mutations of EGFR exons 18–21 in NSCLC patients has been reported [90]. Some that were discovered were not listed in the COSMIC database and not catalogued in cobas or Idylla^TM^ kit, such as L747_A750delinsNRQG, A763_Y764insLQEA, N771delinsHH, D770_N771insGV [91], W817X [92,93], K823E [94], and G857Efs*40 [90]. Furthermore, one of the 1228 patients tested positive for the K823E mutation. It was found that this mutation was able to significantly reduce EGFR activity by decreasing phosphorylation [90,94].

EGFR and its signaling elements are used as targets for the development of new anticancer molecules, such as monoclonal antibodies panitumumab and cetuximab, and tyrosine kinase inhibitors (TKIs), namely gefitinib, erlotinib, afatinib, and osimertinib [95,96,97,98,99]. However, the effectiveness of TKIs has been limited due to the resistance caused by EGFR mutations, and they need further development [98].

## 5. ATP-Binding Site

The ATP-binding site is highly conserved in kinase receptors. Therefore, to develop a more potent inhibitor of EGFR, it is necessary to understand the kinase–inhibitor interaction in the molecular level [99]. Based on structural analysis using several published EGFR tyrosine kinase structures in relation to ligands located at the ATP-binding pocket, 39 residues were found to be close to the binding site and were located in β-sheets, hinge regions, and α-helix. Residues Leu718, Val726, Ala743, Met793, and Leu844 have the most frequent contact in the crystal structures, which are located at β1-3, β-6, and the hinge region. These residues form a core binding pocket that is highly hydrophobic and conserved [62].

Zhao et al. (2019) has grouped the binding modes of all cocrystallized ligands into six clusters by hierarchical cluster analysis. Most of the ligands fall into cluster-1, which showed that the ligands are located at the ATP-binding site and form hydrogen bonds with the amino acids located in the hinge region. The ligands have no interaction with αC-helix. However, the inactive state of EGFR is observed in this cluster, which belongs to the class-6 conformation as described in Section 3. In addition to having a comparable ATP-competitive binding mechanism to cluster-1, the ligand in cluster-3 features a hydrophobic group that penetrates deeply into the hydrophobic pocket at the back of the ATP-binding site, which is often targeted to increase inhibitor selectivity [62]. However, the mutation of the “gatekeeper” residues that is commonly observed in NSCLC patients has a direct effect on the effectiveness of this cluster-3 inhibitor (see Section 4) [60]. The ligands in cluster-4 bind to the hydrophobic and allosteric sites instead of the ATP-binding site, in order not to have a non-ATP-competitive property. Crystal structures of this cluster showed the *out* conformation for the αC-helix, and opened the allosteric site in order for the ligands to interact with the DFG motif in activation loop as well as the αC-helix, and to provide allosteric modulation that has high selectivity to the mutated “gatekeeper” EGFR [100].

The phenyl groups of ligands in cluster-5 interact with the hydrophobic pocket, while the tails interact with aspartate in the DFG-*in* motif, and Asp842 at β-6. This is similar to other ligands positioned in cluster-2. However, their tail interacts with phenylalanine of the DFG-out motif and the G-rich loop of the protein. The furan group of ligands in cluster-6 has different characteristic by forming hydrophobic interactions with the pocket as well as hydrogen bonds with Lys745, Gly762, and the DFG motif [62].

## 6. Allosteric Site

There have been recent developments in compounds targeting the allosteric site in order to produce an alternative medicine that might solve the resistance issue to drugs used in current therapy [101]. Furthermore, the ligands binding to this site, which are also called allosteric modulators of EGFR, lead to conformational changes that may enhance the protein activity and orthosteric ligand binding, or vice versa [102,103,104]. They are divided into three types based on their properties, namely positive (PAM), neutral, and negative allosteric modulators (NAM) [103,105]. These modulators are able to tune the protein kinase activity by interfering with the dynamic interconversion between active and inactive states. In EGFR, the allosteric site of the tyrosine kinase domain is located in the inner pocket of the ATP-binding site. A crystal structure of the EGFR tyrosine kinase domain with an allosteric inhibitor called EAI001 is released into the Protein Data Bank along with the PDB ID of 5D41. Figure 5 shows that EAI001 binds to the allosteric pocket in close proximity to the αC-helix, and is able to inhibit EGFR activation without blocking ATP binding [100,106]. The binding mode of allosteric modulators was also shown along with ligands in cluster-3 and cluster-4, which were mentioned in the previous section.

This binding aims to prevent autophosphorylation and conformational changes of the kinase domain and achieve equilibrium [107]. Therefore, an allosteric modulator can be used to maintain the stability of the interaction between the inhibitor and the kinase ATP-binding pocket. This approach can be applied to solve the drug resistance issue that is often encountered in current treatment because of the inability of TKI inhibitors to bind to mutated EGFR.

## 7. EGFR Tyrosine Kinase Inhibitors

Several advances have been made in recent decades since the identification of activating mutations in the EGFR tyrosine kinase domain in NSCLC patients responding to the first-generation tyrosine kinase inhibitor (TKI) gefitinib, and first-line treatment with EGFR TKIs is now a well-established option in advanced EGFR-mutated NSCLC patients. Various EGFR TKIs have been developed, and some agents have been authorized in these selected patients (Figure 6). Osimertinib, the third-generation EGFR TKI used as the current first-line treatment for NSCLC patients who are EGFR-positive, showed disease-free survival (DFS) improvement in the overall population compared to first-generation EGFR TKIs. Osimertinib underwent phase I trial in 2013 and was approved by the FDA two years later to be used for EGFR T790M-positive NSCLC patients. In 18 April 2018, AstraZeneca announced that osimertinib was approved as the first-line treatment for patients with mutated EGFR (exon 19 deletion or exon 21 L858R substitution) [108]. In contrast, olmutinib, which was developed before Osimertinib, was withdrawn several months after its approval in South Korea due to the occurrence of Stevens–Johnson syndrome and consequent patient death [109].

### 7.1. First-Generation Inhibitors

EGFR was overexpressed in the majority of NSCLC cases. Therefore, treatment was directed towards using inhibitors of this protein kinase. Gefitinib (Iressa^TM^, ZD-1839) was the first EGFR tyrosine kinase inhibitor (TKI) that was approved by the FDA, in 2003, to be used as the first line of treatment for lung cancer with mutated EGFR (Figure 7) [110,111]. This anilinoquinazoline compound selectively inhibits EGFR by reversibly binding to the ATP-binding site and blocking the signal transduction pathways, thus inhibiting cell growth and inducing cell apoptosis [112,113]. Crystal structures of gefitinib and wild-type and mutated EGFR TK have been deposited in RCSB protein databank. These show that gefitinib occupies the ATP-binding site by interacting with several residues in the pocket. The N1 of the quinazoline ring forms a hydrogen bond with Met793 in the hinge region. It also hydrophobically interacts with Leu718, Val726, Lys745, Met766, Leu788, Thr790, and Leu844. Furthermore, gefitinib is classified as a class I inhibitor since it interacts with the active conformation of EGFR with αC-*in* and DFG-*in* [114]. The two randomized phase II trials of gefitinib, IDEAL I and IDEAL II, were conducted as monotherapy in advanced or metastatic NSCLC patients that failed prior chemotherapy treatments. It was found that neither of the trials demonstrated any additional advantage in the survival, progression, or response rates compared to conventional chemotherapy, although they had a more favorable safety profile [113,115,116,117,118]. In the phase III trial (ISEL), gefitinib was able to increase progression-free survival rate, but generally not in advanced-NSCLC patients that had previously undergone chemotherapy [119,120,121,122].

Erlotinib (Tarceva^TM^, CP-358774) was approved by the FDA in 2004 as the first-line treatment of metastatic NSCLC with mutated EGFR, such as deletion in exon 19 and L858R substitution [110]. It is also used as the first line of treatment for NSCLC patients that show progression following at least one chemotherapy regimen. It has the same anilinoquinazoline scaffold as gefitinib but with a symmetrical 2-methoxyethoxy hand in the quinazoline core and an ethyne in the meta position of phenyl group. Furthermore, erlotinib is a type I inhibitor that competes with ATP binding and has similar interaction pattern as gefitinib. It is also a type I½B inhibitor that binds to the EGFR with inactive DFG-*in* and αC-*out* conformation [42,114,123]. A randomized phase II clinical trial using erlotinib alone showed a modest improvement in progression-free survival of none- or light-smoker patients, while a higher rate of adverse events was observed in the use of erlotinib combined with conventional chemotherapy [124]. A placebo-controlled phase III clinical trial in NSCLC patients that had progression after platinum-based therapy demonstrated a longer progression-free and overall survival rate with the use of erlotinib combined with bevacizumab [125,126]. However, a phase II randomized clinical trial showed that there was no superior efficacy of both combinations compared to when using only erlotinib [127].

Since both drugs interact with the gatekeeper residue Thr790, its mutation results in the resistance of gefitinib and erlotinib due to steric clash with aromatic ring [128]. Furthermore, the cell line with L858R/T790M mutation showed a 100-fold lower sensitivity to gefitinib and erlotinib, indicating the resistance of the drugs [76]. The use of gefitinib was restricted in 2005 and withdrawn in 2012 by the US-FDA due to the unfavorable phase III trial results. Therefore, the development of the first-generation TKI is required to increase the effectiveness of NSCLC treatment.

### 7.2. Second-Generation Inhibitors

Anilinoquinazoline derivatives such as afatinib and dacomitinib (Figure 7) were developed to overcome the limitation of the first generation of TKI. Afatinib (Giotrif^TM^, BIBW-2992) is an irreversible inhibitor of EGFR with exon-19 deletion and L858R substitution, which causes resistance to gefitinib and erlotinib [129,130]. The two crystal structures of afatinib in complex with wild-type and T790M EGFR were deposited in the protein databank with ID 4G5J and 4G5P, respectively [131]. Similar to gefitinib, afatinib forms hydrogen bonds with the backbone of Met793 in the hinge region. It also interacts with the hydrophobic region in the same way as gefitinib. The furanyl group was exposed to the solvent and the 3-chloro-4-fluorophenyl group was located close to the gatekeeper residue. It was classified as a type VI inhibitor because its acrylamide group binds covalently to the Cys797 of the active conformation of EGFR [114]. Furthermore, two randomized phase II clinical trials were conducted to compare afatinib and platinum-doublet chemotherapy in patients with advanced lung carcinoma harboring mutated EGFR. Afatinib showed longer progression-free survival compared to the chemotherapy, although there was no significant difference in terms of overall survival [132,133,134]. Based on the pooled analysis of the overall survival data for patients with exon 19 deletion of EGFR, there was a significant increase in the median overall survival by 31.7 months compared to chemotherapy. However, there was no benefit in the analysis for patients with L858R EGFR. These data indicate that afatinib is the first EGFR TKI to show overall survival benefit compared to chemotherapy in patients with EGFR exon 19 deletion [135].

Dacomitinib (Vizimpro^TM^, PF-299804) is also an irreversible EGFR TKI that has been approved by the FDA to be used for patients with metastatic NSCLC with exon 19 deletion and exon 21 substitution [136,137,138,139]. It shares similar binding properties with afatinib and forms hydrogen bonds with the hinge residue and hydrophobic interactions with those in the binding pocket. It is also a type VI inhibitor because the acrylamide group covalently interacts with Cys797 [114]. This irreversibly covalent interaction is the characteristic of second-generation TKIs. Resistance to dacomitinib was found in EGFR with C797S substitution [140,141]. Furthermore, a randomized phase III clinical trial (ARCHER 1050) of dacomitinib versus gefitinib was carried out on NSCLC patients with mutated EGFR. It was found that dacomitinib has more promising results in terms of progression-free survival compared to gefitinib. However, more serious adverse effects were observed in patients that were given dacomitinib [142].

### 7.3. Third-Generation Inhibitors

The third-generation TKIs, namely osimertinib and rociletinib (Figure 7) were developed to overcome the resistance of EGFR secondary mutation and to reduce the cases of serious adverse effects. Osimertinib (Tagrisso^TM^, AZD-9291) is the first TKI with a non-quinazoline core approved by the FDA to be used as the adjuvant therapy for NSCLC patients that have undergone resection and have EGFR with mutations in exon 19 and 21 [143]. It is an irreversible TKI that is used for the EGFR T790M mutation, but has less activity against wild-type EGFR [143,144,145]. The deposition of the crystal structures of osimertinib along with wild-type and mutated EGFR in the databank in good resolution [146,147] have made it possible to carry out interaction studies on osimertinib and he EGFR TK domain. Furthermore, due to the crystalline structures, the nitrogen in the pyrimidine ring is able to establish a hydrogen bond with the N-H backbone of Met793. This indicates that the pyrimidine is able to mimic the interaction of the quinazoline core of other TKIs in the binding site. The acrylamide group covalently interacts with Cys797, thus it is classified as type VI inhibitor [114]. The hydrophobic interactions of osimertinib in the binding pocket were similar to those of other TKIs. However, the acquired EGFR mutation (C797S substitution) causes resistance to this drug [141]. A double-blind phase III clinical trial (FLAURA) showed that osimertinib gave a significantly longer progression-free survival than other TKIs in advanced-NSCLC patients with previously untreated mutated EGFR [148]. It also has similar safety profile to other TKIs but with lower rates of grade 3 or higher adverse events, which suggests its superiority to the other TKIs [133,148,149,150,151].

Rociletinib (Xegafri^TM^, CO-1686) is an irreversible inhibitor of mutated EGFR. According to preclinical and phase I/II clinical studies, this drug has minimal efficacy against wild-type and exon-20-insertion EGF [152,153]. Furthermore, the crystal structures of rociletinib in complex with EGFR T790M and L858R were also studied [154]. It was found that in the T790M EGFR, the anilinopyrimidine group in rociletinib forms two hydrogen bonds with Met793 amide and carbonyl backbone. Rociletinib was also able to form two hydrogen bonds in EGFR L858R. These include one between nitrogens in the pyrimidine group, and another between the fluoromethyl and Thr790, which became a hydrophobic interaction in the T790M structure. The acrylamide group in rociletinib covalently binds to Cys797 in both active (DFG-*in*/αC-*in*) conformation structures [154]. In addition, a randomized phase III trial (TIGER-3) showed that rociletinib has longer progression-free survival but higher rates of hyperglycemia compared to chemotherapy in advanced-NSCLC patients with mutated EGFR (excluding exon 20 insertion) [155].

### 7.4. Fourth-Generation Inhibitors

It was reported that EGFR mutation C797S causes resistance to irreversible second and third TKIs [140,141,156,157]. In order to solve this issue, a large library of compounds targeting EGFR L858R/T790M was screened, and two compounds were obtained as the next-generation TKIs, namely EAI001 and EAI045 (Figure 7) [100,158]. These compounds have high selectivity towards EGFR L858R/T790M compared to the wildtype. Furthermore, the high resolution of the EGFR T790M/V948R crystal structure in complex with EAI001 that was deposited in the databank had an allosteric binding property with the ligands. This indicates that EAI001 is a selective allosteric inhibitor of mutant EGFR [100]. Instead of binding near the hinge residues, EAI001 binds to the deeper pocket and forms a hydrogen bond with Asp855 in the allosteric site of EGFR (Figure 5). The mutation of Thr790 into methionine enables the thiazole group of EAI001 to form a pi-sulfur interaction with the mutated gatekeeper residue. The aromatic rings have hydrophobic interactions with Met766, Leu777, Leu788, and Phe856, that may have a key role in the allosteric activity. Furthermore, the binding of the allosteric inhibitor prevents autophosphorylation of kinase and stabilizes protein conformation [107].

The optimization of EAI001 resulted in the formation of EAI045, which is a highly selective allosteric inhibitor of EGFR L858R/T790M. The binding mechanism of EGFR T790M/C797S/V948R with EAI045 was disclosed in the 2.90 crystal structure of this ligand in the presence of the C797S substitution, which causes resistance to type 1 TKIs [158]. Furthermore, EAI045 is able to form more polar interactions than the former compound EAI001. It forms three hydrogen bonds, namely carbonyl oxygen of the isoindolin-1-one to Lys745, amide nitrogen to Asp855, and para-fluorophenol to Phe856 in the allosteric binding pocket, and hydrophobic interactions with Met766, Leu777, Leu788, and Phe856 side chain. These interactions lead to better affinity and solubility of the compound. However, the EAI045 interaction with Lys745 appears to be impossible in the actual condition. In the EGFR–EAI001 complex, Lys745 interacts with the negatively charged AMP-PNP rather than the ligand. This might be due to the strong polar interaction that “pulled” the side chain of Lys745 away from EAI001, causing it to preferably interact with the AMP-PNP rather than the allosteric ligand. Furthermore, due to the fact that the ATP molecule is missing from the crystal structure of EGFR T790M/C797S/V948R-EAI045, the binding affinity of these two complexes cannot be compared equally [158].

The in vitro assay showed that the quantity of EAI045 reduced but did not fully eliminate EGFR autophosphorylation or kinase function in NSCLC and an H1975 cell line with EGFR L858R/T790M. The efficacy of EAI045 was also evaluated and then was used in combination with cetuximab in an in vivo study using a mouse model with L858R/T790M mutant-driven lung cancer. The mice treated with EAI045 and cetuximab showed remarkable tumor regression, while the one that was only given EAI045 showed no response. The same result was also found in L858R/T790M/C797S-engineered Ba/F3 cells as well as mice harboring L858R/T790M/C797S tumor xenografts, which demonstrated that EAI045 is superior to other TKIs. It was also able to overcome acquired resistance to T790M and C797S mutations [159]. Therefore, clinical trials are required to confirm its effectiveness in advanced-lung-cancer patients.

### 7.5. Other Allosteric Inhibitors

A high-throughput docking was performed using the mutated EGFR–EAI045 complex, a L858R/T790M mutant EGFR, and a homology model of EGFR, which produced 92 small molecules that were predicted to have allosteric behavior with the kinase protein. These compounds were further screened using in vitro assay, and compound N-((2-methyl-1H-indol-3-yl)(pyridin-2-yl)methyl)-3-nitroaniline (refer to compound 8 in the publication) was obtained as the most promising allosteric inhibitor that is able to interact with wild-type, T790M, and L858R/T790M EGFR [160]. Furthermore, in order to determine the effect of the stereochemistry, two enantiomers of compound 8 were docked to the protein. The nitrophenylalanine of (S)-enantiomer formed a hydrogen bond with the carboxyl group of Asp855 in the allosteric binding site. It also had hydrophobic interaction with several residues in the pocket. In addition, the pyridine and indole group were flipped in the (R)-enantiomer to enable it form three hydrogen bonds, including a polar contact with Asp855. The pyridine ring of (R)-enantiomer formed a hydrogen bond with Lys745, while the indole group interacted with Phe856. This latter enantiomer interacts hydrophobically with residues that are similar to those found in the (S)-enantiomer. Therefore, this indicates that both enantiomers are capable of interacting favorably inside the allosteric region. The compound was able to inhibit the activity of wild-type and L858R/T790M double-mutation EGFR in 4.7 μM and 6.2 μM, respectively. The dose–response curve did not change when the concentration of ATP varied. This indicates that the compound inhibits the double-mutant EGFR via an allosteric mechanism. In addition, according to a ligand-based similarity analysis, compound 8 differs structurally from the allosteric inhibitors EAI001 and EAI045 [160].

### 7.6. Small Molecules in the Clinical Trials

Several novel TKIs have been developed to address resistance issues caused by both common and rare EGFR mutationss. Table 1 lists the drugs which are now undergoing clinical trials, followed by the condition being studied.

## 8. Future Direction

The current generation of EGFR TK inhibitors has experienced drawbacks, and most notable is the issue of mutant EGFR resistance. A number of common and rare mutations discovered in NSCLC patients diminish the sensitivity to TKIs given as therapy, and lower overall survival rate. Furthermore, drug safety must be considered. Olmutinib, which had been previously approved by the Republic of Korea in May 2015, had to be withdrawn several months later due to Stevens–Johnson syndrome as its adverse reaction [109].

Therefore, more powerful and efficient inhibitors are required. The availability of crystal structures for both native and mutant EGFR has promoted the development of NSCLC medicines targeting EGFR. EAI045 was designed based on the optimization of current inhibitors used in therapy, although the in vivo finding revealed that EAI045 was only beneficial when coupled with cetuximab [161]. This compound needs to be further developed in the future to solve the problem of resistance, while maintaining the inhibitory activity of the molecule. One of the strategies is to retain the structure or group capable of forming interactions with the key amino acids in the kinase binding site. The other possible strategy is combining the TKI with other agents that can overcome the various resistant tumor clone.

Apart from the development of TK domain inhibitors, the extracellular region of the EGFR, which is known as the binding site for native ligands EGF [21] and the monoclonal antibodies panitumumab and cetuximab [161], may also be targeted in the development of NSCLC treatments. Understanding the important interactions as well as the appropriate peptide length for extracellular EGFR can also help in the development of new inhibitory compounds in this region [90]. The development of novel compounds capable of binding to the rare mutant EGFR would be an intriguing research area. Therefore, a greater understanding of EGFR activation and dimerization mechanisms may aid in the development of new TK inhibitors in the future.

## Figures and Tables

**Figure 1 molecules-27-00819-f001:**
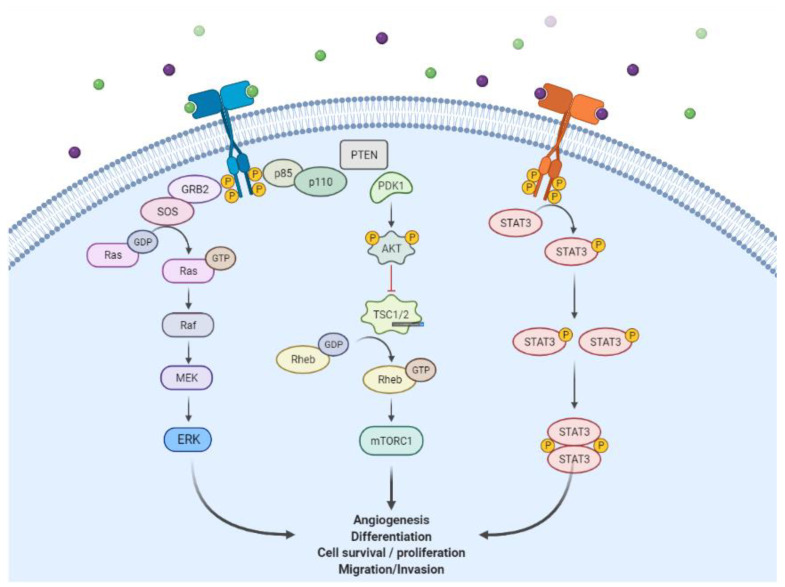
HER/ErbB signaling (created by BioRender.com on 14 October 2021). Green and purple spheres indicate the EGFR-activating ligands.

**Figure 2 molecules-27-00819-f002:**
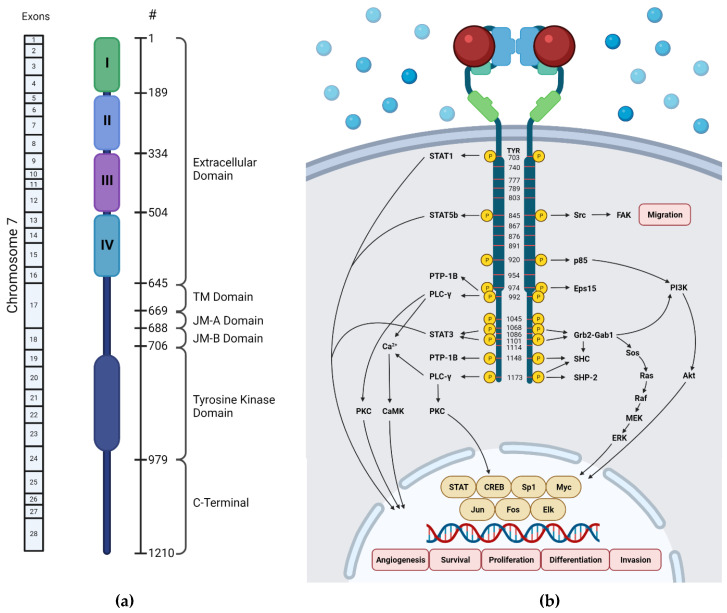
Schematic diagrams of EGFR domains. (**a**) Domain structure of human EGFR and exons encoding it (created by BioRender.com on 14 October 2021), (**b**) EGFR phosphorylation sites [18]. Blue spheres indicate the molecules present outside the cell, and red spheres indicate the EGFR-activating ligand.

**Figure 3 molecules-27-00819-f003:**
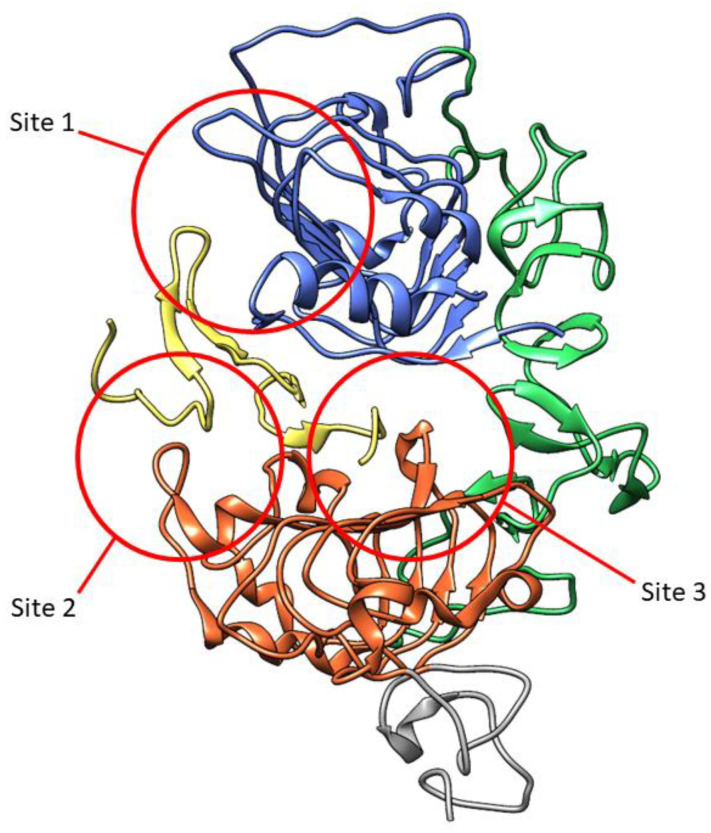
Three-dimensional visualization of EGFR extracellular domain in complex with EGF (yellow) (PDB ID: 1IVO). Subdomains are marked in colors: L1, blue; CR1, green; L2, orange; part of CR2, grey. The three sites interacting with EGF are marked in red circles.

**Figure 4 molecules-27-00819-f004:**
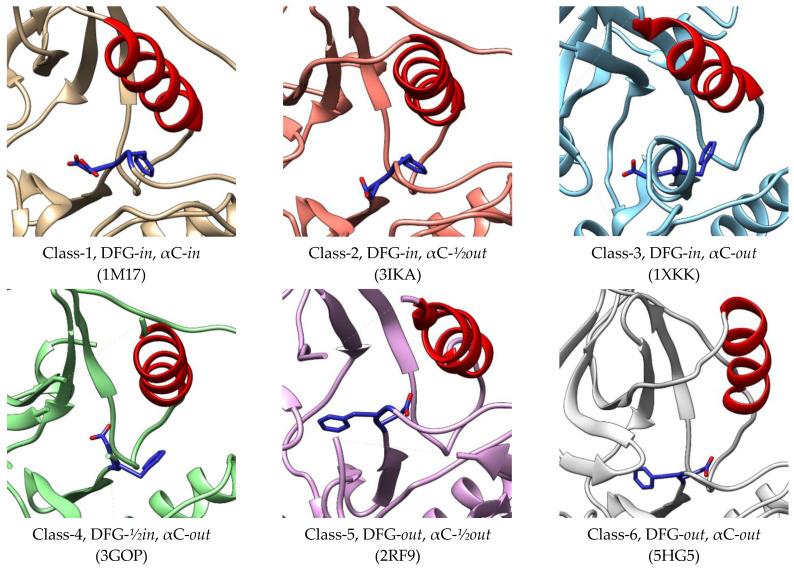
Conformations of active and inactive EGFR classified by Zhao et al. (2019) [62]. The crystal structures 1M17 [42] and 3IKA [63] represent class-1 and class-2 of the active conformations, while 1XKK [64], 3GOP [30], 2RF9 [65], and 5HG5 [66] represent class-3, class-4, class-5, and class-6 of the inactive conformations, respectively.

**Figure 5 molecules-27-00819-f005:**
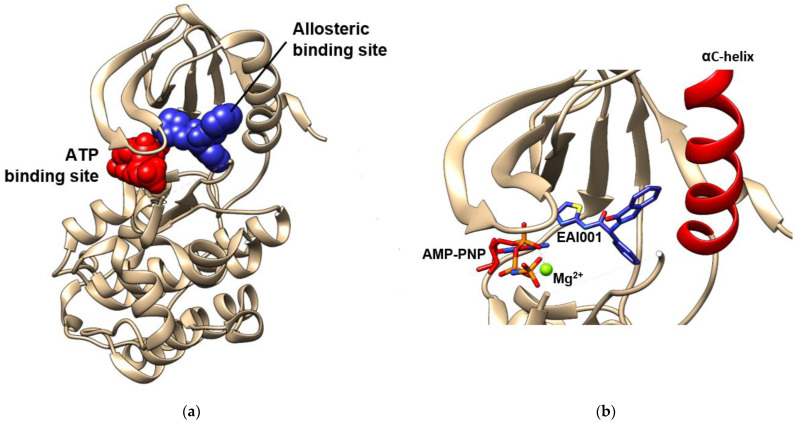
ATP and allosteric binding site of EGFR TK domain. (**a**) Allosteric binding site is marked by blue spheres, while ATP-binding site is marked by red spheres. (**b**) The close visualization of the allosteric inhibitor EAI001 and AMP-PNP in the binding pocket. EAI001 binds to the allosteric site close to αC-helix. The visualizations are made using the crystal structure with PDB code 5D41 [100] by Chimera 1.15 (accessed on 13 August 2021).

**Figure 6 molecules-27-00819-f006:**
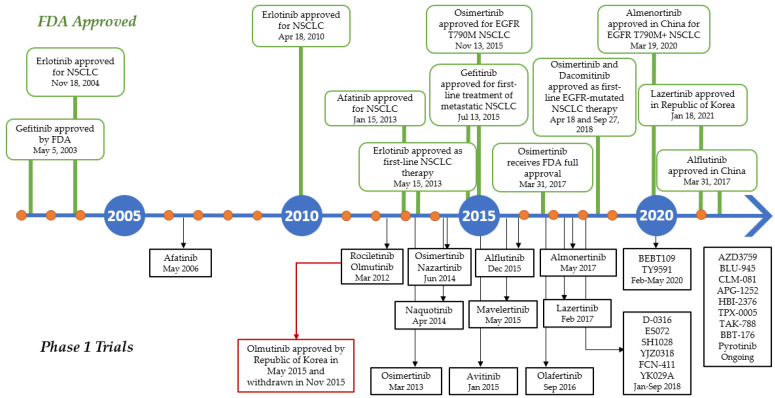
Timeline in the development of EGFR TKIs during the last two decades. Green boxes above the line indicates the drugs approved by FDA, while the black boxes below the lines indicates the phase I clinical trial of the TKIs.

**Figure 7 molecules-27-00819-f007:**
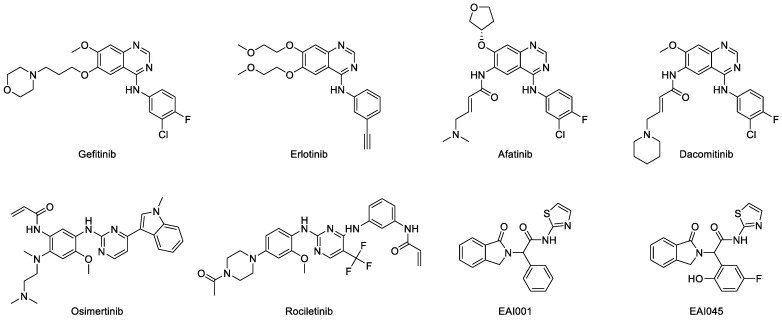
Chemical structure of EGFR TKI (first to fourth generations).

**Table 1 molecules-27-00819-t001:** Investigational TKIs in the pipeline.

Drug	Clinical Trial Identifier ^a^	Phase	Condition ^c^
Lazertinib (YH25448)	NCT03046992	I/II	NSCLC
	NCT04075396	I/II	NSCLC
	NCT05167851	II	NSCLC
	NCT04248829	III	NSCLC
D-0316	NCT03452150	I	NSCLC
	NCT04206072	II/III	NSCLC
	NCT03861156	II	NSCLC, ST
AZD3759	NCT03360929	I/II	NSCLC
	NCT03653546	II/III	NSCLC, BM
FCN-411	NCT03420079	I	NSCLC
DZD9008	NCT03974022	I/II	NSCLC
Nazartinib (EGF816)	NCT03333343	I	NSCLC
	NCT02108964	I/II	NSCLC
	NCT03292133	II	NSCLC
Icotinib	NCT05007938	II	NSCLC
	NCT03749213	II	NSCLC
	NCT03396185	II	NSCLC
	NCT03349203	II	NSCLC
	NCT02737774	II	AC
Apatinib	NCT03801200	II	NSCLC
	NCT04824352	II	OS
	NCT03913182	II	EC
	NCT04253873	II	HGG
	NCT03475589	IV	GC, NSCLC. BC, OC
BLU-945	NCT04862780	I/II	NSCLC
Avitinib (AC0010)	NCT02330367	I/II	NSCLC
	NCT03574402	II	NSCLC
Almonertinib (HS-10296)	NCT04905550	II	NSCLC
	NCT04952168	II	NSCLC
	NCT04785742	II	NSCLC
	NCT04636593	II	NSCLC
	NCT04685070	III	AC
CLN-081	NCT04036682	I/IIa	NSCLC
APG-1252	NCT04210037	I	SCLC
	NCT04893759	Ib	NT
	NCT04001777 ^b^	Ib	NSCLC
	NCT05186012	Ib/II	NHL
	NCT04354727	Ib/II	MF
Furmonertinib	NCT05079022	I/II	AC
	NCT04858958	Ib	NSCLC
	NCT04982900	II	LC
	NCT04965831	II	AC
	NCT04970693	II	NSCLC
	NCT04895930 ^b^	II	NSCLC
	NCT04853342	III	NSCLC
HBI-2376	NCT05163028	I	NSCLC, CC, PC, ST, PC
ABT-414	NCT02573324	III	GS
Dasatinib (BMS-354825)	NCT02954523	I/II	NSCLC
	NCT00529763	II	LK
	NCT01471106	II	BC
Repotrectinib (TPX-0005)	NCT05004116 ^b^	I	ST
	NCT04772235	I	NSCLC
	NCT03093116	I/II	ST
	NCT04094610	I/II	ST
	NCT05071183 ^b^	Ib/II	ST
Poziotinib (NOV120101)	NCT03066206	II	NSCLC
	NCT03744715	II	NSCLC, BC
	NCT03066206	II	NSCLC
	NCT03318939	II	NSCLC
	NCT04172597	II	BC, CC, ST, HGG
Larotinib (Z650)	NCT04131192	Ib	PC
	NCT03888092	Ib	EC
	NCT04415853	III	EC
Mobocertinib (TAK-788)	NCT04056455	I	HV, RI
	NCT04056468	I	HV, HI
	NCT04051827	I	NSCLC
	NCT03807778	I/II	NSCLC
	NCT04129502	III	NSCLC
Vandetanib (ZD6474)	NCT00537095	II	TC
	NCT00410761	III	TC
	NCT01876784	III	TC
	NCT00418886	III	NSCLC
BBT-176	NCT04820023	I/II	NSCLC
Brigatinib (AP26113)	NCT04634110	II	NSCLC, BM
	NCT04223596	II	NSCLC
	NCT03535740	II	NSCLC
	NCT04074993	II	NSCLC
	NCT03596866	III	NSCLC
Pyrotinib	NCT04680091	I	HV
	NCT00600496	I	BC, CC, LC, KC
	NCT04960943	II	GT
	NCT04380012	II	CC
	NCT04646759	III	BC

^a^ ClinicalTrials.gov identifier. ^b^ in combination with other anticancer therapies. ^c^ AC, adenocarcinoma; BC, breast cancer; BM, brain metastases; CC, colorectal cancer; EC, esophageal cancer; GC, gastric cancer; GS, gliosarcoma; GT, gastrointestinal tumor; HGG, high-grade glioma; HI, hepatic impairment; HV, healthy volunteer; KC, kidney cancer; LC, lung cancer; LK, leukimia; MF, myelofibrosis; NHL, non-Hodgkin lymphoma; NT, neuroendocrine tumor; NSCLC, non-small-cell lung cancer; OC, ovarian cancer; OS, osteosarcoma; PC, pancreatic cancer; RI, renal impairment; SCLC, small cell lung cancer; ST, solid tumor; TC, thyroid cancer.

## Data Availability

Not available.

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
