# Peer review of "Structural Insight and Development of EGFR Tyrosine Kinase Inhibitors"

_molecules, 2022, doi:10.3390/molecules27030819_

Round 1
Reviewer 1 Report
The review manuscript entitled “Structural Insight and Development of EGFR Tyrosine Kinase Inhibitors" is worth publishing in the Journal. However, there are some queries which need to be addressed before accepting the manuscript. Following are my observations;
- The manuscript needs English editing.
- Provide a table related to the investigational drugs in the pipe-line.
- Future direction should highlight the limitations of clinical development of newer agents as to why more and more drugs are failing in terms of clinical utility.
- Provide a timeline scheme for the development of EGFR Tyrosine Kinase Inhibitors.
- As many of the approved EGFR Tyrosine Kinase Inhibitors have inspired the design and discovery of several newer agents in the preclinical stages. Appending these will improve the quality of review and help in engaging more readers.
Reviewer 2 Report
This review by Daryono Hadi Tjahjono and colleagues reported structural insight and development of EGFR tyrosine kinase inhibitors in a detailed manner. The authors nicely reported the structural insight for EGFR and its inhibitors. Overall, it’s a nice piece of work. However, I suggest some minor modifications:
- Summarized in table form comparison of different generation approved and in pipeline drugs binding mode conformation, clinical trial stage, approval status, etc.
- Check referencing: Some of the paragraphs cited with multiple referencing but the content mostly relates to one article. To avoid readers' confusion it is advisable to cite a specific article. For example line 290 cited 62,64,103 but the author discussed mostly 64
Reviewer 3 Report
To address resistance to mutated EGFR, more potent and potent inhibitors are needed. The current generation of EGFR TK inhibitors are deficient, and the availability of natural and mutant EGFR crystal structures has facilitated the development of EGFR-targeting NSCLC drugs. One such strategy is to maintain structures or groups capable of interacting with key amino acids in the kinase binding site. Another possible strategy is to combine TKIs with other drugs that can overcome various drug-resistant tumor clones. EAI045 was designed based on optimization of current inhibitors used in therapy. The compound needs to be further developed in the future to address drug resistance while maintaining the inhibitory activity of the molecule. The development of new compounds capable of binding to rare mutant EGFR would be an interesting area of research. This review is therefore very suitable for publication in the Molecules journal.
